# Engaging Stakeholders to Solve Complex Environmental Problems Using the Example of Micropollutants

**Thomas Hillenbrand** [1,*], **Felix Tettenborn** [1], **Marcus Bloser** [2], **Stephan Luther** [3], **Adolf Eisenträger** [4], **Janek Kubelt** [4] and **Jörg Rechenberg** [5]

1    Fraunhofer Institute for Systems and Innovation Research ISI, Breslauer Straße 48, 79139 Karlsruhe, Germany; felix.tettenborn@isi.fraunhofer.de
2    IKU GmbH, Olpe 39, 44135 Dortmund, Germany; bloser@dialoggestalter.de
3    Federal Ministry for the Environment, Nature Conservation, Nuclear Safety and Consumer Protection (BMUV), Robert-Schuman-Platz 3, 53175 Bonn, Germany; stephan.luther@bmuv.bund.de
4    German Centre for Micropollutants (SZB), Wörlitzer Platz 1, 06844 Dessau-Roßlau, Germany; adolf.eisentraeger@uba.de (A.E.); janek.kubelt@uba.de (J.K.)
5    German Environment Agency (UBA), Wörlitzer Platz 1, 06844 Dessau-Roßlau, Germany; joerg.rechenberg@uba.de
*    Correspondence: thomas.hillenbrand@isi.fraunhofer.de

**Abstract:** Current and future challenges such as the climate crisis, demographic change and achieving the objectives of the Water Framework Directive require holistic and precautionary approaches within the framework of national and supranational strategies. Specific measures and projects resulting from these strategic activities are required to successfully meet the challenges. In 2016, the German Environment Agency (UBA) and the Federal Ministry for the Environment (BMUV) commissioned the process for the development of the Federal Government's micropollutants strategy, which was later named the Trace Substance Strategy. The essential core instrument herein was a multi-stakeholder dialogue aimed at giving sufficient consideration to the different interests of the various stakeholders. The goal was to develop a balanced mix of measures and to initiate implementations in order to reduce emissions of micropollutants as effectively and efficiently as possible, at the source, in their application and in the downstream areas. The various measures were tested in a pilot phase, and the activities were evaluated before being transferred into the subsequent consolidation phase. This article describes the outcomes of the stakeholder dialogue as an instrument. This is complemented by the results of a stakeholder evaluation of the process itself and the results achieved. Important outcomes of the stakeholder dialogue are a Committee for the Identification of Relevant Micropollutants and the use of roundtables as an important instrument in which the manufacturers and the users of the substances can make an important contribution to reducing emissions. To address the opportunities and necessities of additional wastewater treatment, an "orientation framework" for municipal wastewater treatment plants was also established. Additionally, the German Centre for Micropollutants (SZB) was founded to continuously organize, support and accompany the various outcomes that became relevant pillars of the German government's Trace Substance Strategy. The evaluation has shown that new approaches and new instruments have been created within the framework of the stakeholder dialogue, which enable flexible and short-term options for action and allow for the involvement of stakeholders in a manner appropriate to the polluter-pays principle. Specific emission reductions could not be expected within the time frame of the dialogue. However, stakeholders agreed that the strategic process chosen is preferable to purely regulatory steps due to the holistic approach involving all stakeholders concerned in this complex matter. It is expected that in the future, if implemented consistently, the approach could achieve a lasting and sustainable impact on a broad scale.

**Keywords:** micropollutants; trace substances; strategy; open government partnership; stakeholder engagement; stakeholder dialogue; corporate responsibility; measures for the reduction of micropollutant emissions; polluter-pays principle; precautionary principle

## 1. Introduction

Pollution of waters with chemicals is a national and global environmental problem that is addressed by various Sustainable Development Goals, especially goal 6.3 (By 2030, improve water quality by reducing pollution, eliminating dumping and minimizing release of hazardous chemicals and materials) [1,2]. Although specific environmental policy measures and requirements for improving water quality have been established at European and national levels for many years [3–5], the need for action remains very high. Micropollutants, which are referred to as trace substances in the context of the German multi-stakeholder dialogue, can have adverse effects on aquatic ecosystems even at low concentrations (see e.g., [6,7]) and affect the production of drinking water (see e.g., [8–11]). The accumulation of poorly degradable substances, by-products that may be formed or even the effects of the complex mixtures of substances that result in the environment are additional causes for concern [5,12]. Micropollutants originate from different sectors and sources and enter the environment via a variety of pathways (Figure 1). Accordingly, different starting points for measures for reducing micropollutant emissions as well as the combination of measures are conceivable and necessary, and their implementation is to be addressed and driven through a comprehensive strategy at national and/or, if necessary, regional level. In principle, options for action of a technical and non-technical nature are available for all emission points, which means that the implementation of measures can be started at the various starting points. Due to the different sectors and sources within the complex emission scheme of micropollutants, various sector-specific policies and regulations are to be considered.

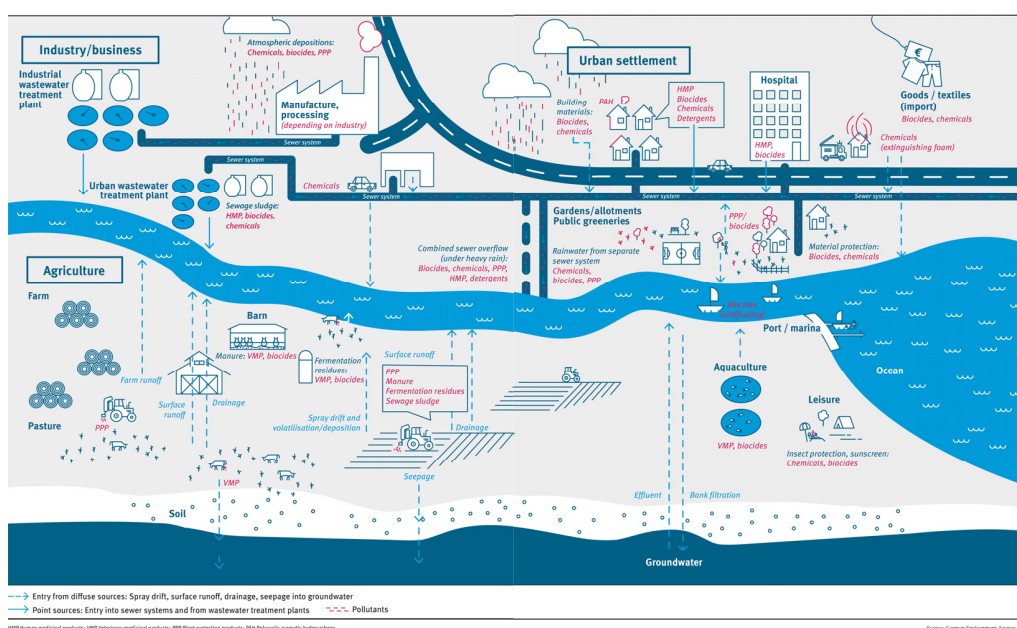

**Figure 1.** Schematic overview of possible entry pathways of micropollutants into waters, published in [5]. Reprinted with permission from German Environment Agency, 2023, Jörg Rechenberg.

The different variables—various pollutants, environmental conditions and stakeholder groups—the dynamics with which awareness of the issue is increasing and how emissions of different substances are evolving over time, the uncertainties regarding the impacts of some substances and even more of different cocktails of diverse substances present in water bodies, as well as information gaps e.g., on the exact amounts of substances emitted from different sectors, and last but not least, especially the conflicts of interests and goals from the different stakeholder groups, including their relationship with each other (e.g., environmental protection vs. economic efficiency) contribute to a situation that could be described as a "wicked problem" (e.g., [13,14]).

For a holistic and precautionary solution, policy mixes need to be established that target information gaps and integrate the specifics of the different sectors [15]. In 2015, the

German Environment Agency (Umweltbundesamt, UBA) published a study on "measures to reduce micropollutant emissions to water." The primary objective of this study was to develop suitable and highly cost-effective measures or combinations of measures and general conditions to reduce the emission of micropollutants into water via the municipal wastewater system [16].

In 2015, based on these results and inspired by the pragmatic approach in Switzerland, the German Environment Agency launched a position paper entitled "Organic micropollutants in waters—fourth treatment stage for less pollution" [17]. This paper recommended updating the best available technology in wastewater treatment and the introduction of more advanced wastewater treatment processes (fourth treatment stage) in municipal wastewater treatment plants (WWTPs) of a specific size (>100,000 inhabitant equivalents) as well as smaller WWTPs that discharge into sensitive water bodies. These recommendations were complemented by a second study in 2016 [18] which produced more specific cost and efficiency data for an advanced fourth treatment stage at municipal wastewater treatment plants. Additionally, the cost contribution over different plant sizes and the overall economic benefit of the emission reduction measures were assessed. One of the recommendations of this study was the proposal of a holistic strategy, integrating all relevant stakeholders targeting risk characterization, risk management and risk communication [18].

In a workshop discussing these results in Berlin (2016), representatives of municipal wastewater associations confirmed that in their eyes a solely "end-of-the-pipe-approach," as they called the fourth treatment stage, was not a comprehensive solution for the diverse inputs of micropollutants into water. The representatives of the German Federal Ministry for the Environment, Nature Conservation, Building and Nuclear Safety (Bundesministerium für Umwelt, Naturschutz, nukleare Sicherheit und Verbraucherschutz, BMUV) declared that they also would prefer a holistic approach that addresses all sources of micropollutants into water bodies. In this approach, they wanted to integrate the whole variety of instruments for reducing the inputs of micropollutants, beginning by raising awareness via voluntary measures and ending with potential legal and economic measures. Based on the understanding of the necessity of integrating the various stakeholders involved in the emissions of micropollutants, this was the starting point for a stakeholder dialogue bringing together the views of all relevant interest groups on this issue, which is meanwhile in accordance with the fundamental principle of governance for sustainability worldwide [19–23].

The aim of this article is to describe the method of the multi-stakeholder dialogue applied here as an instrument for developing the German government's Trace Substance Strategy to enable the necessary policy mix, taking into account the various sectors involved. The achieved results and still open aspects are discussed and assigned to the findings of the recent literature. The discussion of the outcomes of the dialogue process is complemented by a reflection of the results of a stakeholder evaluation of the process itself and the results achieved.

## 2. Methods

### 2.1. Multi-Stakeholder Dialogue as an Instrument

The introductory remarks (Section 1) already indicate the very complex initial situation that exists in the subject area of limiting environmental pollution by micropollutants. The most important general conditions that must be taken into account when developing solutions are:

- The large number of relevant substances and substance groups
- The very different environmental effects, which are also only known to a limited extent, for example with regard to their combination effects, and whose assessment is therefore subject to uncertainties
- The different areas of production, industrial use, application and entry paths

- The large number of actors involved and responsible for substance and product manufacturing and application, as well as for wastewater treatment and water protection measures, along with the large number of different areas of chemicals and water regulation.

With regard to the various fields of action, it must be taken into account that relevant regulations already exist that relate to individual aspects with regard to the recording of environmental pollution with pollutants and their reduction. Wagner [24] shows that the precautionary principle (preventing hazards to the environment from occurring in the first place) and the polluter-pays principle (oblige the party responsible for the pollution to pay the damages) according to the EU Treaty (Art 191, para. 2 TFEU) and a risk-based approach, which is of paramount importance in the REACH Regulation 1907/2006 or comparable regulations for human and veterinary medicinal products, biocides or plant protection products (approval of specific applications if the risk to health or the environment is adequately controlled) can neutralize each other. Ahting et al. [5] emphasize the importance of the precautionary principle and justify this in particular based on the uncertain risks associated with micropollutants. Metz and Ingold [25] also argue similarly. In contrast, the EU's Environmental Quality Standards Directive (2008/105/EC) only sets requirements for the environmental status of selected pollutants (priority substances and certain other pollutants listed in Annex 1 of the abovementioned directive). At the same time, it obliges the member states to ensure compliance with the requirements by means of their own measures.

However, the evidence of pollution with micropollutants both in surface waters and in groundwater shows that these regulations are not sufficient for a comprehensive solution to the problem. The evidence of pollution is available both at the national level in Germany [5,26] and at the international level [27].

As Schaub et al. [15] described, policy mixes mostly evolve incrementally over a long time period without a unifying goal, which "can be conflicting and produce counterproductive effects, leading to unsustainable outcomes." As a possible approach for overcoming this problem, Schaub et al. [15] pointed out the power balance between stakeholders as an example. In this sense, in addition to the already existing regulations, a very comprehensive approach has to be adopted, which does more than merely provide for reactive measures when a risk has already been proven or monitoring values have shown an environmental impact. Rather, it is important to proactively identify and prevent negative developments in advance.

A variety of approaches can be used (see [16,18,28]). The present work showed that to develop the necessary comprehensive approach, "sectoral policies" are not sufficient; instead, cross-sectoral perspectives and approaches are needed that aim well beyond existing regulations. Therefore, combinations of specific individual means are necessary and, at the same time, "established sectoral communication structures" must be overcome.

As an approach to develop a comprehensive strategy at the national level, the BMUV and UBA therefore launched a dialogue process in which all relevant stakeholders were involved. The following points were seen as key advantages of such a stakeholder dialogue (cf. [29,30]):

- Such a dialogue process makes it possible to involve all relevant stakeholders from the very beginning.
- Through the joint technical discussions, a common understanding of the problem can be achieved, even if the initial situations and general conditions differ significantly among the various groups of actors.
- Solutions developed jointly in a discourse also receive high levels of acceptance in terms of further implementation.
- A joint dialogue with the government and administration means that those involved are aware of political decisions at an early stage and can adapt to future developments at an early stage, for example, in product developments or investment decisions.

- If the dialogue process is successful, improvements can be achieved in the short term, for example, through voluntary measures that can be implemented quickly by the stakeholders.
- A dialogue process also provides opportunities to identify problems that cannot be solved within the framework of the process or starting points or measures for which no consensus can be reached, and to highlight the additional need for political action that may be associated with them.

However, in order to achieve jointly supported results within the setting of such a dialogue process, all relevant stakeholders must be prepared to participate, i.e., the pressure to act must have been recognized in principle. In addition, good general conditions must be created for the process and various principles for success must be observed. Following the Harvard concept [31] and analyses of stakeholder dialogues in the environmental sector based on it [32], the following aspects in particular were taken into account:

- Separation of factual and relationship levels, supported by professional moderation
- High levels of transparency both within the process (interests of the stakeholders, processed, for example, in the course of preliminary discussions with the stakeholders; processing of internal information and materials) and outside the process (documentation and publication of agreed results)
- Competence: Technical support of the process to identify and describe new starting points and solutions, with additional involvement of external experts, if necessary, e.g., via the UBA
- Efficiency of the process: Ensuring the most efficient processing possible, e.g., by delegating detailed work to separate working groups and providing professional support for the overall process
- Legitimacy: Especially at the beginning, but also in the further course of the process, clarification of the mandate and the internal legitimacy for the individual stakeholder groups in order to ensure a binding nature of the solutions to be developed
- Fairness and mutual trust: Ensuring fair and appreciative interaction with each other through jointly developed rules and compliance with them via independent moderation; protected, non-public space for building trust within the framework of the dialogue.

For a description of the implementation of these aspects within the framework of the dialogue, see Section 2.2.

### 2.2. Stakeholder Dialogue: Process Design

The main objective of the process of the stakeholder dialogue was the development of contributions to a national strategy on trace substances of the federal government.

### 2.2.1. Preparation Phase

To implement the process, a project team consisting of Fraunhofer ISI (scientific support and processing of technical issues) and IKU GmbH (moderation and organizational support) was commissioned to carry out the process in close coordination with BMUV and UBA. The process began in June 2016, lasted through several phases and was formally completed in June 2022 (Figure 2); the main results are documented in Hillenbrand et al. [33]. The formal framework of the process was characterized by the following:

- Preparatory interest analysis based on exploratory talks with target groups of the dialogue (cross-section of authorities, manufacturing and application companies and actors, trade associations, environmental and consumer protection associations and technical experts from the fields of action pharmaceuticals, plant protection, biocides, detergents/cleaning agents, cosmetics, industrial cleaning agents and surface treatment).
- Identify stakeholders' concerns and interests and their potential contributions to achieve potential objectives (good water status, precautionary principle and state of the art/best available technology)

- Development of a catalog of fundamentally suitable measures for dealing with micropollutants on the basis of the expert discussions and supplementary literature evaluations. The list of suggestions served, among other things, as a basis for the selection of stakeholders. The stakeholders were also to be given the opportunity during the exploratory phase to contribute their own proposals for measures and to name taboo topics (measures that, from the point of view of individual stakeholders, are not negotiable).
- Involvement of all relevant stakeholders representing those responsible for causing or solving the problem, associations from the fields of civil society and environmental protection, as well as the federal states responsible for reducing pollution in administrative enforcement (business associations, health sector, environmental and consumer protection, companies, water management, municipalities, involvement of other authorities). In addition, the process was accompanied by other federal ministries.
- Establishing clear rules for communicating (interim) results and agreeing on proposed measures. The aim was to achieve a trusting approach to proposals and impact assessments as quickly as possible as part of the stakeholder dialogue.
- Monitoring of the process with strict separation of roles between Fraunhofer ISI (scientific support, background papers, technical support for work assignments and working groups) and IKU (process management in dialogue).

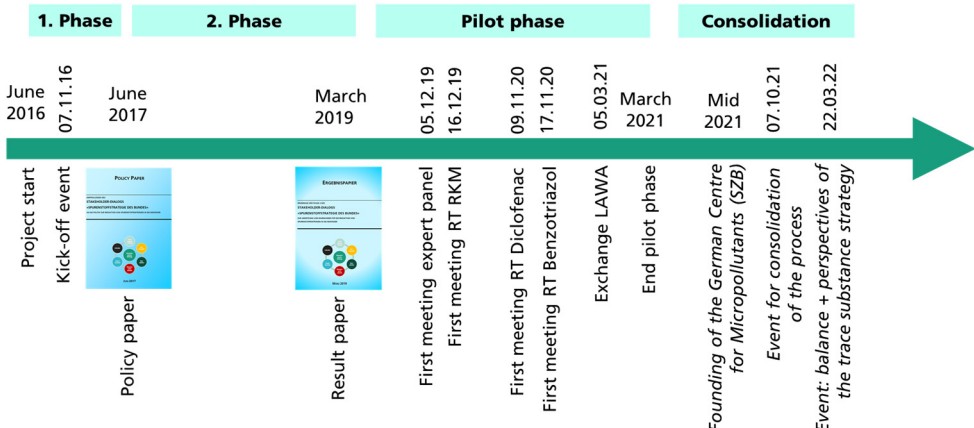

**Figure 2.** Overview of the dialogue process on the German federal government's Trace Substance Strategy. Source: translated according to [33].

The approximately 30 stakeholders included covered the different groups shown in Figure 3, involving the following:

- The sector of water distribution and wastewater management via the representative German water associations ('Water res. man.' in Figure 3)
- Producers and professional users of substances that can lead to micropollutants if they are emitted into water bodies (human pharmaceuticals, biocides, crop protection products, industrial and household chemicals, personal care and laundry products) and products that contain said substances
- Local authorities with responsibility for the control of substance emissions and the German Working Group on Water Issues (Bund/Länder-Arbeitsgemeinschaft Wasser, LAWA) of the Federal States and the federal government represented by the Federal Environment Ministry
- NGOs as representatives for the environment and civil society ('Civil society' in Figure 3).



**Figure 3.** Relevant stakeholder groups in the context of micropollutants. Source: Fraunhofer ISI.

The complete list of involved stakeholders can be found in the Appendix A. In the selection of stakeholders, a balanced mix of the various stakeholder groups was sought, ensuring that sufficient direct participation in a possible solution to the problem was ensured via stakeholder involvement. The selection was regularly discussed with the stakeholders in the run-up to and during the process and, if necessary, adjusted by means of additions.

From the beginning, the role of the German Environment Agency within the stakeholder dialogue was under discussion. On the one hand, with their broad experiences on micropollutants, evaluation of their findings in water bodies, technologies to remove them and instruments to reduce them at the source or downstream, representatives of the Agency contributed essential information on different topics within the dialogue. On the other hand, the Agency made clear that it was not a stakeholder within the process and would not be part of the formal decision process on results and products (policy papers) of the dialogue. Thus, the Agency would not be bound by decisions of the stakeholders. This enabled the Agency to keep its scientific independence and to evaluate the results of the stakeholder dialogue during the entire process. As its legal task is to give advice to the Federal Ministry of Environment, the Agency published its own recommendations for reducing micropollutants in waters in 2018 to give scientific input to the dialogue [5]. From these, it can be seen that in the view of the Agency, voluntary agreements alone will not be enough to reduce the emission of micropollutants into waters to a sufficient degree. Therefore, the Agency proposes to take additional legal and economic measures.

### 2.2.2. Phase 1

In the first phase of the dialogue process, three workshops were held on the most important areas for action in the life cycle of micropollutants:

- Mitigation strategies at the sources (January 2017)
- Mitigation strategies in application (February 2017)
- Possibilities of downstream ("end-of-pipe") measures (March 2017)
- An additional workshop was held in May 2017 to consolidate the results

Figure 4 shows the three areas of possible starting points for mitigation measures. The first two topics, i.e., source- and application-oriented approaches, are particularly closely linked.

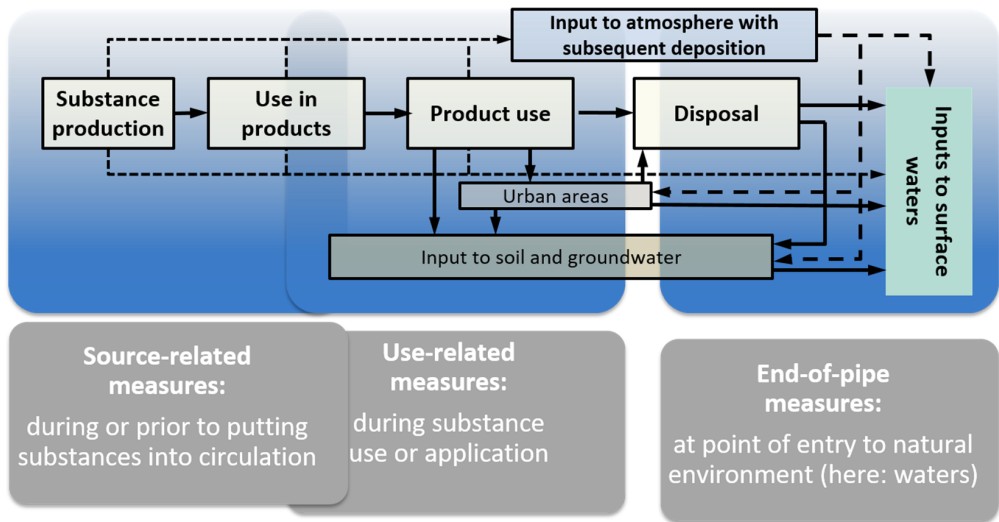

**Figure 4.** Starting points for emission reduction measures. Source: modified according to [18].

Results of the first phase were documented in the policy paper "Recommendations from the stakeholder dialogue on the Trace Substance Strategy of the German federal government to policy-makers on options to reduce trace substance inputs to the aquatic environment," which contains 14 recommendations in total. Figure 5 shows the broad range of recommendations across the different areas of action. For the elaboration of these measures, a common understanding of the problem of unwanted micropollutants in the aquatic environment was required as a basis. Additionally, there was a desire among stakeholders to further participate in the development of measures and their implementation.

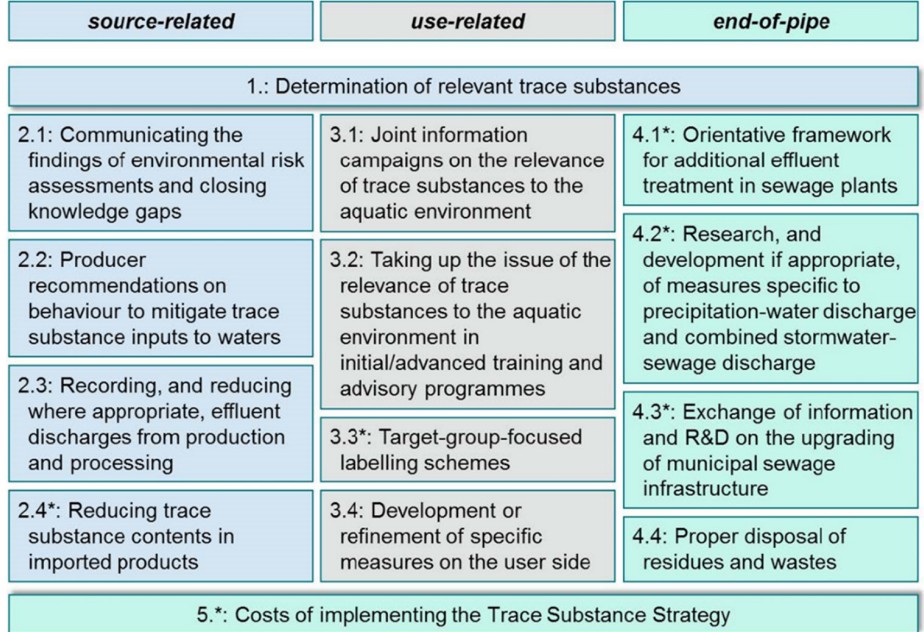

**Figure 5.** Results of the first phase of the stakeholder dialogue of the federal Trace Substance Strategy, documented in [34]. * Recommendations on which dissenting opinions were voiced by individual stakeholders. These opinions are stated in full in their respective contexts in [34].

2.2.3. Phase 2

In order to enable a continuation of the dialogue process as requested by the stakeholders and a specification of the 14 recommendations for action of the policy paper with the active participation of the stakeholders, the process was continued with a second phase.

A new collaborating structure was implemented for the process, which is shown in Figure 6. The group of stakeholders established in the first phase of the project was transformed into a "Stakeholder Forum" with slight modifications. This forum formed the umbrella for the implementation of the recommendations with the task of steering the overall process and defining the work program and framework (composition of the forum, rules of cooperation, etc.). Under the umbrella and on behalf of the Stakeholder Forum, various working groups were formed; these were responsible for the specification of selected recommendations from the policy paper.

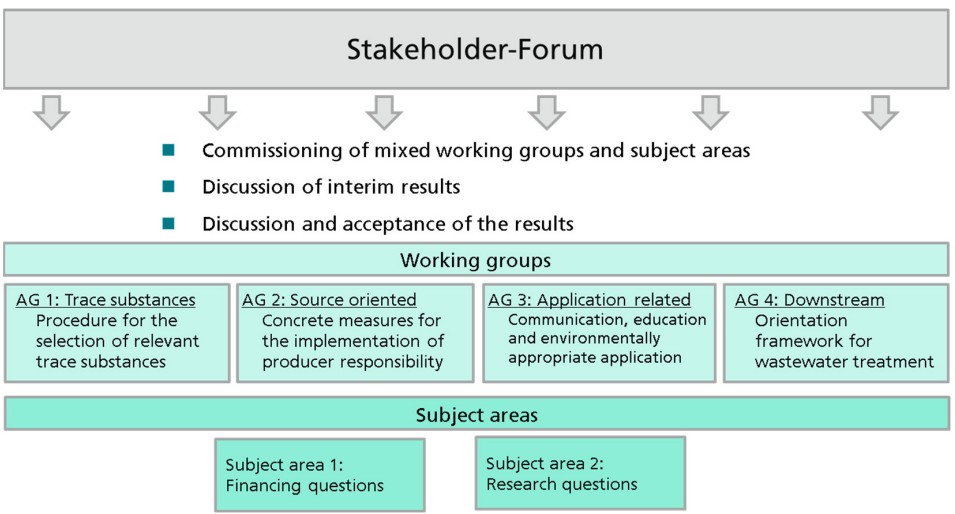

**Figure 6.** Stakeholder dialogue "Federal Strategy on Trace Substances"—overview of the working structure of Phase 2. Source: Fraunhofer ISI.

The main results from the second phase of this stakeholder approach were 15 measures in total (Figure 6), which were more specific than in phase 1. Once again, these were clustered into source-oriented, application-oriented and downstream measures (Figure 7). An overarching measure additionally includes a procedure for identifying relevant micropollutants.

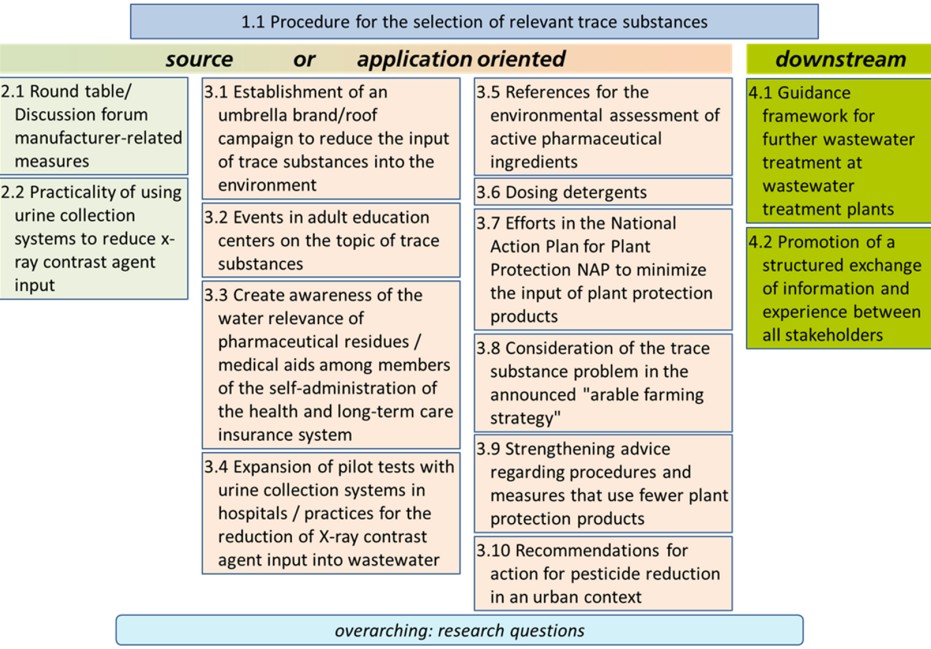

**Figure 7.** Overview of results of the second phase of the stakeholder dialogue of the federal Trace Substance Strategy. Source: [35].

### 2.2.4. Pilot Phase

For further specification and implementation, a pilot phase was launched in 2019 following phase 2 of the dialogue process. Selected measures and activities from the work within the framework of the process were taken into account and continued. The German Centre for Micropollutants was founded to institutionalize and firmly anchor the work. The focus was on the following activities:

- Preparations were made for the establishment of the German Centre for Micropollutants at the UBA to ensure the exchange of information and experience between the actors and to be able to support further steps and activities via a central coordinating body.
- An expert panel was established to issue final decisions on the relevance of micropollutants that are assessed by the UBA.
- The instrument of the roundtables was used to identify relevant prevention and reduction measures for specific substances or groups of substances within the framework of producer responsibility and to initiate their implementation. Roundtables were initiated for a total of three micropollutants/micropollutant groups.
- The developed "orientation framework" for advanced wastewater treatment was tested by the federal states.
- In order to achieve a common communication strategy, the BMUV prepared the consolidation of activities under the umbrella of the UN Water Decade (2018–2028) (https://wateractiondecade.org/, last accessed on 24 March 2022).
- The various activities of the different dialogue phases that have taken place up to this point were assessed as part of a comprehensive evaluation by the stakeholders involved in the development and implementation of the activities. A particular focus was placed on the entire process as well as on the expectations regarding the achievement of objectives and the evaluation of the results achieved.

### 2.2.5. Consolidation Phase

After completion of the pilot phase, the German Centre for Micropollutants (Spurenstoffzentrum des Bundes, SZB) was founded in 2021 with the objective of assuming a coordinating and integrating role in the field of "micropollutants in water bodies." In order to ensure that the various activities could continue as smoothly as possible during the establishment phase of the new Centre, the continuation phase ran until mid-2022.

### *2.3. Evaluation*

Because final evaluations of the results and the various measures and instruments cannot be made due to the delayed effects of the environmental improvements to be achieved, the focus was therefore not only on an assessment of the achievable results of the measures, but also on an evaluation of the overall process and its sub-areas.

The development of the evaluations was essentially based on interviews with the stakeholders involved in the overall process and in the individual processes with the aim of further improving the measures that have been started, to give input and address expectations and desires to the newly built German Centre for Micropollutants (SZB), and to optimize future processes.

The evaluation took place in several, successive steps:

- End of 2020: Evaluation surveys (online) of the participants of the Round Table RKM, as it was the first roundtable to start, as well as of the Panel for the Assessment of the Relevance of Trace Substances
- February 2021: Comprehensive survey (online) of the participants of the different dialogue phases as well as of the members of the roundtables to evaluate the interim status of the process achieved at that time (with differentiation between process and result evaluation)
- March 2021: Presentation and discussion of the results as part of an event on the results of the pilot phase

- Continuation in 2022: Since only partial results of the roundtables were available at the beginning of 2021 and the work of the roundtables was continued very intensively in the further course of 2021, the evaluation of this instrument was extended. For this purpose, a workshop with representatives of all stakeholder groups and all roundtables was held on 4 March 2022. The results were presented and discussed at the review event on 22 March 2022.

The online surveys were conducted based on EFS survey by Tivian XI GmbH. A total of 70 people took part in the comprehensive survey in February 2021. The various groups of actors were fully covered:

- Federal states/environmental administration
- Manufacturers of relevant products (including business associations)
- Environmental associations and civil society
- Water management
- Others, including professional users, municipalities/communities (including municipal top associations), science/research.

The results of closed questions were analyzed quantitatively, while feedback on open questions was analyzed qualitatively. The results were presented to the stakeholders within a broad presentation, allowing them to give feedback on the results of the survey and on the process.

## 3. Results

### 3.1. Results of the Dialogue Process

The stakeholder dialogue led to many results that built on the proposals for measures that were made in the first and second phase of the dialogue, which were fleshed out in the pilot and continuation phases and have already been implemented in some cases. Some important implementations are described in more detail below.

- Implementation of The German Centre for Micropollutants (SZB):

In the pilot phase, the objectives and topics of the Centre were specified with the involvement of stakeholder feedback. The main objective of the Centre is the continuation of the activities within the framework of the Trace Substance Strategy, e.g., the supervision of the panel for the evaluation of the relevance of trace substances or the initiation and support of roundtables. The Centre has successively started its work in 2021. In order to ensure the smoothest possible continuation of the previous work of the federal Trace Substance Strategy, the work was transferred to the Centre as part of a transition phase.

- Implementation of an expert panel for the final decision on the relevance of micropollutants after assessment of the UBA:

The background to the relevance assessment panel is the need for a simplified procedure for substance assessment in order to be able to prioritize the derivation of measures at the national level within the Trace Substance Strategy—e.g., for the initiation of roundtables for individual trace substances or trace substance groups. Therefore, in the second phase of the dialogue, all stakeholders agreed on specific criteria for a rather quick and simple assessment of the relevance of micropollutants for substances to be further addressed within the strategy, but not with the aim of replacing national or European regulations.

- Installation of roundtables for selected micropollutants to engage responsibility of industry:

A total of three roundtables were initiated on the following trace substances or trace substance groups:

- X-ray contrast agents (Röntgen-Kontrastmittel, RKM): start date, December 2019
- Diclofenac: start date, November 2020
- Benzotriazole: start date, November 2020.

Measures to mitigate water pollution for the respective substances were developed by these roundtables, which were largely adopted by consensus of the stakeholders involved. Table 1 provides an overview.

**Table 1.** Overview of the most important results of the roundtables.

| | X-ray Contrast Agents (RKM) | Diclofenac | Benzotriazole |
|---|---|---|---|
| **Addressing information deficits** | Significance of emissions during production/processing Feasibility of the urine collection concept ($\rightarrow$ Study) | Need for awareness-raising among doctors, pharmacists, users Emissions with topical application ($\rightarrow$ Study) | Test procedure Stiftung Warentest Relevance of different application areas (Symposium, Cooling lubricant $\rightarrow$ Study) |
| **Measures** | (Measures during production/processing) Awareness-raising measures Pilot projects: broad application of the urine collection concept | Information measures regarding topical application ("wiping instead of washing") Publications and awareness-raising activities Pilot studies with project advisory board | Further development of test procedures (with the involvement of manufacturers) until end of 2022 General awareness-raising measures Targeted information in application sectors |
| **Implementation** | Partly already being implemented, pilot projects from the end of 2022 | Partly already being implemented, three pilot studies from 2022/2023 onward | Partly already being implemented |

Source: [33].

- Information campaign(s) under the umbrella of the UN Water Action Decade:

During the pilot phase, the possibility was created to bring together projects and campaigns initiated by stakeholders on the topic of micropollutants under a common umbrella and to support them by awarding them the national UN Water Decade logo (https://www.bmuv.de/themen/wasser-ressourcen-abfall/binnengewaesser/un-wasserdekade, accessed on 24 March 2021). To accompany these processes, overarching requirements for stakeholder projects or campaigns were developed that had to be met when using the Decade logo. A campaign on the correct disposal of pharmaceuticals was also carried out by the Federal Ministry of Environment. The logo was also awarded to other initiatives and projects.

- Application of the "orientation framework" for advanced wastewater treatment in the federal states:

Water pollution with micropollutants can be significantly reduced by advanced wastewater treatment in municipal wastewater treatment plants. Within an "orientation framework", criteria were listed which play an important role in the prioritization of plants with regard to expansion, the pollution situation of the water bodies concerned, efficiency criteria, utilization requirements or the sensitivity of the water bodies. The application of this framework developed in phase 2 was formally recommended to the federal states by the German Working Group on water issues of the Federal States and the Federal Government represented by the Federal Environment Ministry (LAWA). The feedback from the federal states showed that their situations were heterogeneous due to the different starting points and the different assessments of the affectedness and urgency in the states. The application of the framework is considered to be a time-consuming process, and the approaches and experiences with implementation have varied so far. An important

aspect is the need to improve the financial framework for implementation (see below). The implementation and the professional discussions around this topic should be continued in the future.

- Funding:

After only a short period, the participants of the stakeholder dialogue realized that when the debates came down to the funding of measures, agreements could not be reached. This was partly due to the fact that the representatives of most of the participating associations had no mandate from their members to make binding financial concessions. However, the participants also realized that organizing and implementing financial instruments as charges on certain products containing micropollutants or obligatory measures as additional wastewater treatment are ultimately governmental tasks which could not be decided within the stakeholder dialogue. Therefore, the participants decided that the issue of funding should be dealt with outside of the dialogue. The BMUV agreed with this view. Nevertheless, to take on board the different views on the issue of funding, the Federal Ministry of Environment and the UBA organized two events where scientists, lawyers, economists and representatives of authorities from Germany and abroad gave their views on existing and potential funding instruments. As expected, no agreement could be reached between the stakeholders, but the events gave valuable input into the political debate on economic instruments for giving incentives for the reduction of micropollutants into water bodies and their legal restrictions. The participants agreed that at least the following criteria should be considered when conceptualizing funding instruments: effectiveness, efficiency, constitutional and European law restrictions, practicability, enforceability, transaction costs, controllability and distributive justice. The political debate was accompanied and fueled by a diverse range of studies commissioned by participants of the stakeholder dialogue. This shows the great significance of this issue on the one hand and on the other hand how the stakeholder dialogue was and is a source for scientific debate and studies.

- Open Government Partnership (OGP):

On 30 June 2021, the German Government passed the Third National Action Plan (NAP) 2021–2023 in the framework of Germany's participation in the Open Government Partnership (OGP). This action plan contains 11 commitments by the Federal Government or its agencies. These include measures for improving access to federal legal information as well as open data from the areas of integrity management and public procurement. Under top 6.8 of this action plan, the German Government commits itself to "Maintaining the dialogue on trace substances," meaning that the process that emerged from the pilot dialogue on trace substances is to be continued. The admission to the NAP shows a great appreciation for the work done by the stakeholders within the dialogue. It realizes that the dialogue offers maximum transparency and takes a cross-sectoral approach. It follows that this is an obligation to continue the dialogue with a view to producing specific measures and broadening it to include further target groups.

*3.2. Results of Evaluation by the Engaged Stakeholders*

The evaluation considered both process-related aspects and the results achieved. In principle, the high level of complexity, the very different starting points in the various areas of application and regulation, and the large number of stakeholders with widely differing interests must also be taken into account—aspects that were reflected, among other things, in the high level of effort required to carry out the process. The level of effort was also rated as high by some individual stakeholders, while others considered it to be appropriate.

The survey also asked about satisfaction with individual measures. For example, the objectives of the expert panel in the pilot phase (relevance assessment of micropollutants as a basis for initiating measures, as well as application and further development of the method for assessing relevant trace substances within the framework of the federal Trace Substance Strategy) were slightly more than "partially" achieved according to the stakeholders' assessment (mean value 4.4 out of 7). In this assessment, there were very clear differences

between the various stakeholder groups (Figure 8): The highest rating came from the federal states/environmental administration group (5.22), while the lowest came from the product manufacturers (3.71) and the environmental associations and civil society (3.83) groups. These last two groups often have polarizing positions on environmental regulation, which is not surprising given the potential consequences that "relevant" labeling of a particular substance can have on the manufacturers' side and given the higher expectations of the environmental associations and civil society in a faster process, covering more substances or substance groups, that could be handled during the pilot phase.

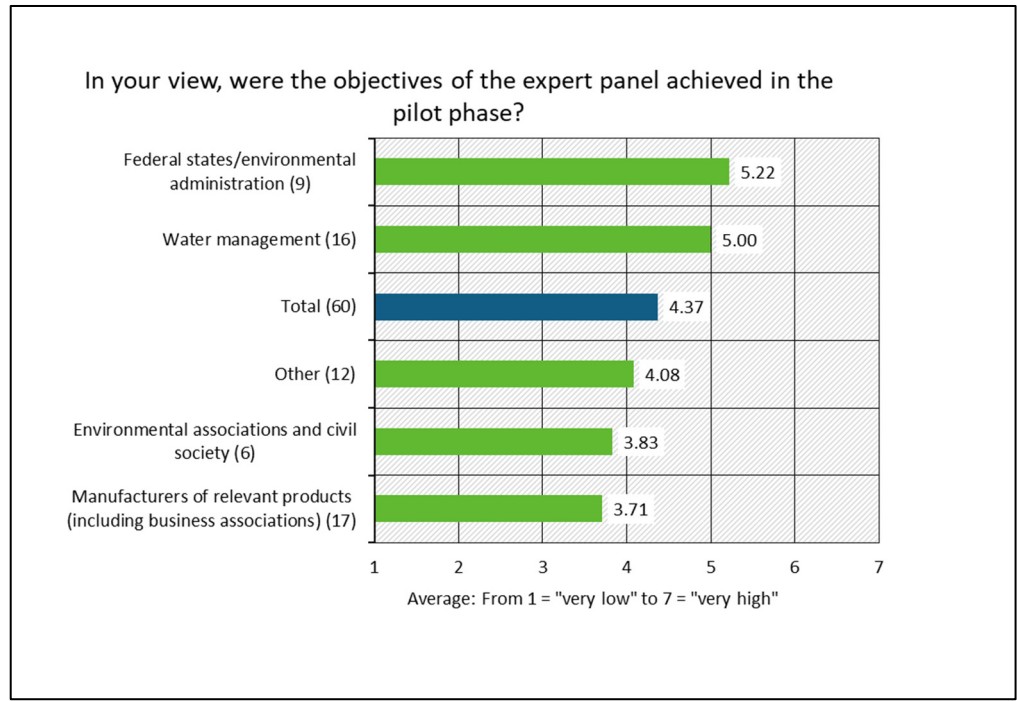

**Figure 8.** Survey results: achievement of the expert panel's objectives, differentiated by stakeholder group. Source: own illustration, Fraunhofer ISI.

With regard to the roundtable instrument, stakeholders were asked about the chances of success in using this instrument to (a) develop efficient measures for individual micropollutants and substance groups and (b) initiate their implementation. The results show that the chances of success for (a) are assessed as medium to high (mean score: 4.6 out of 7) and for (b) as slightly higher than medium (mean 4.2 out of 7). Differentiated by stakeholder groups, product manufacturers (5.41 and 5.29 out of 7, respectively) and the federal states/environmental administration group (4.89 and 4.0 out of 7, respectively) show a significantly more positive assessment than water management (3.81 and 3.25 out of 7, respectively) and environmental associations/civil society (each 4.0 out of 7; Figure 9a,b). As environmental associations and civil society and water management normally prefer a strong state to just take regulatory action, as e.g., Schaub & Tosun [36] explained in their positioning of environmental groups in a stakeholder dialogue, the rather low rated values from these two groups are not surprising.

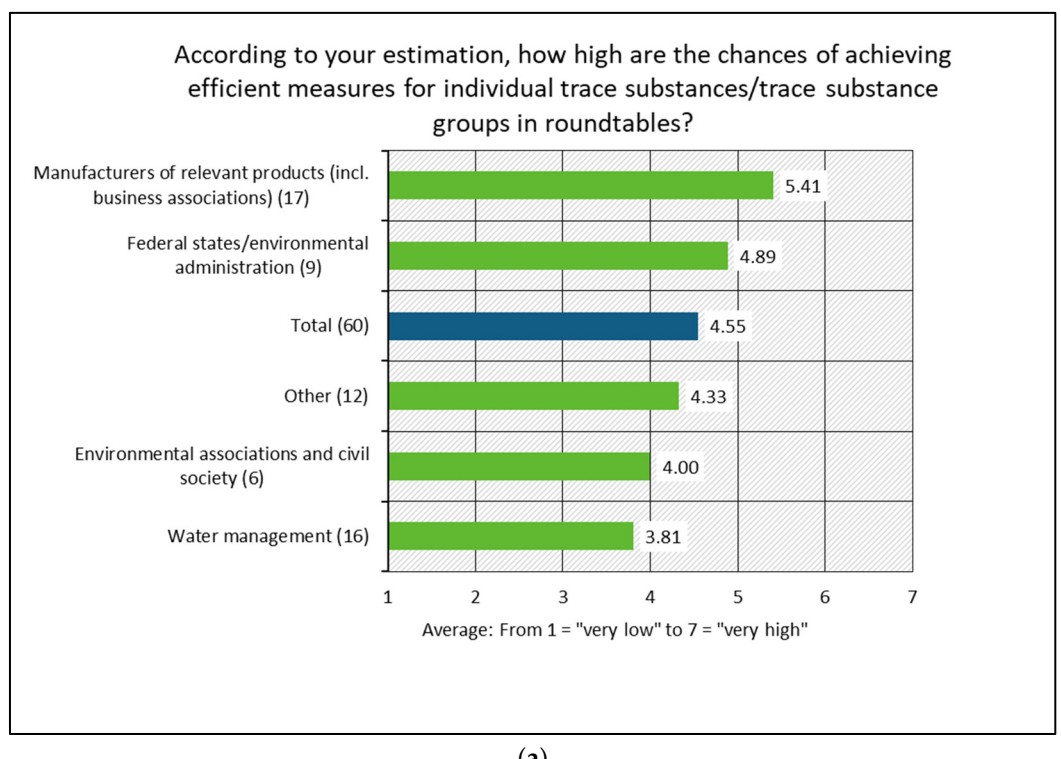

(**a**)

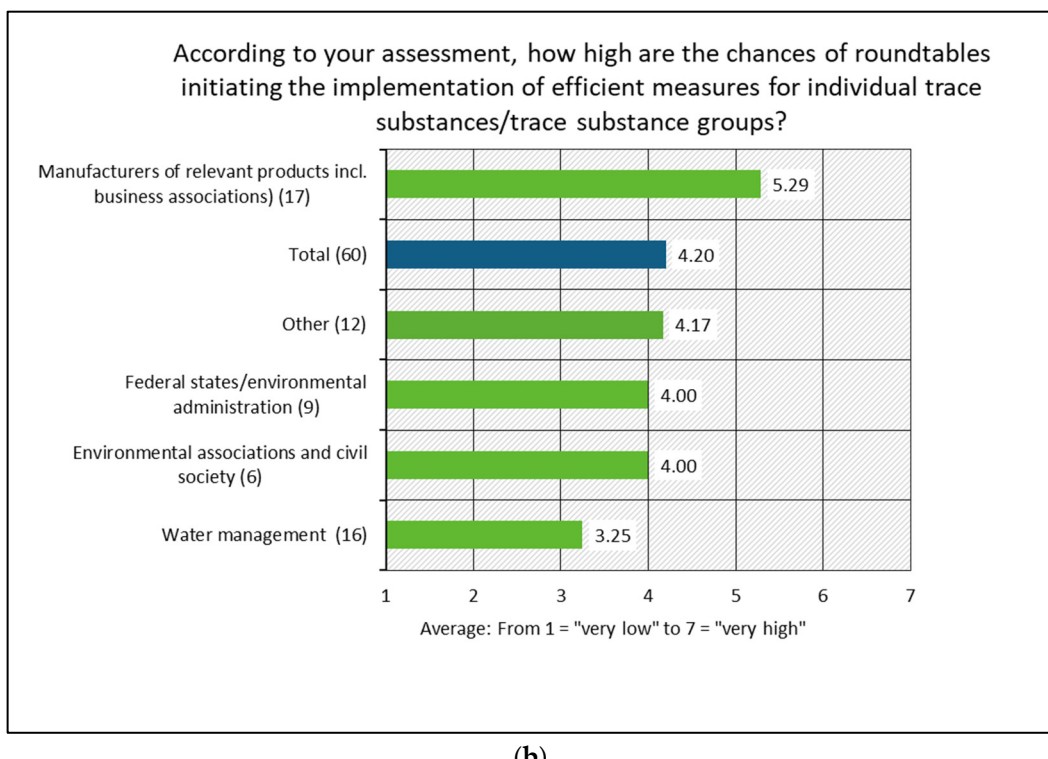

(**b**)

**Figure 9.** Survey results—opportunities for roundtables to (**a**) develop and (**b**) implement measures, differentiated by stakeholder groups. Source: own illustration, Fraunhofer ISI.

The effort associated with roundtables is rated as very high. However, the associated effort varies among the different stakeholder groups and is rated as difficult by some of them, particularly among the environmental associations/civil society group. On the basis of a clear organizational framework with a specified, efficient working structure and good technical preparation, the time frame of such a roundtable should therefore be limited

to about one year. Whether it makes sense to initiate a roundtable for a micropollutant or substance group depends on the specific application and emission situation of the micropollutant or trace substance group. In this sense, the following questions must be clarified in advance:

- What is the relevance of current inputs into the environment or water bodies? To what extent are there information gaps that can be closed through cooperation between different stakeholders?
- To what extent do existing regulations already limit inputs to the environment or water bodies? Are currently existing gaps known?
- Are options for action known and how can they be assessed in terms of their effectiveness and feasibility? Are there opportunities to exert influence through information or awareness-raising measures, for example? Or are there known, specific obstacles to implementation that could be removed in the context of a roundtable?

The survey results for the "orientation framework" show that the success of the application is rated as moderately successful with a mean score of 3.6 out of 7 (Figure 10). Differentiated by stakeholder groups, the highest rating is shown by product manufacturers and by federal states/environmental administration group, and the lowest by water management.

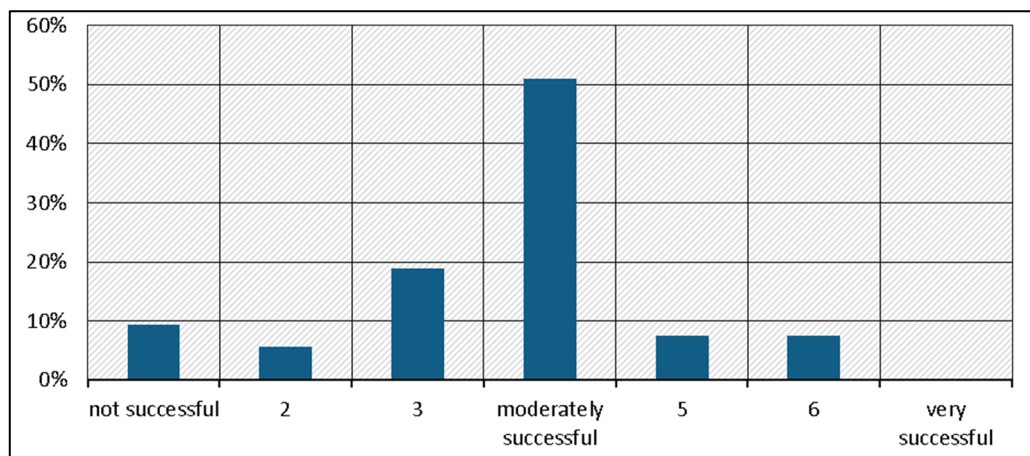

**Figure 10.** Survey results—success of the application of the "orientation framework." The graph is based on N = 53 responses. Source: Fraunhofer ISI.

Stakeholders were satisfied with the chosen approach of using a dialogue process involving stakeholders to develop the basic principles and important elements of the federal Trace Substance Strategy (mean score: 4.9 out of 7, cf. Figure 11). The comprehensive participation of stakeholders (mean score: 5.80 out of 7), the promotion of a broad understanding (mean score: 5.51 out of 7) and the development of a broad approach to solutions (mean score: 5.37 out of 7) were seen as particular advantages in this regard. Since the measures implemented in the pilot phase required varying degrees of stakeholder involvement, not all stakeholders were equally involved in all measures implemented (e.g., orientation framework, information campaigns). An additional aspect for the somewhat lower relevance that was given to the balance of measures (mean score: 4.83 out of 7, cf. Figure 12) was probably that an important aspect of the process was the development of source- and use-related measures to fulfill the idea of the precautionary polluter-pays principle in addition to regulatory steps. From a holistic perspective, this approach was complemented by upstream and downstream measures (especially assessment of the relevance of substances and orientation frameworks).

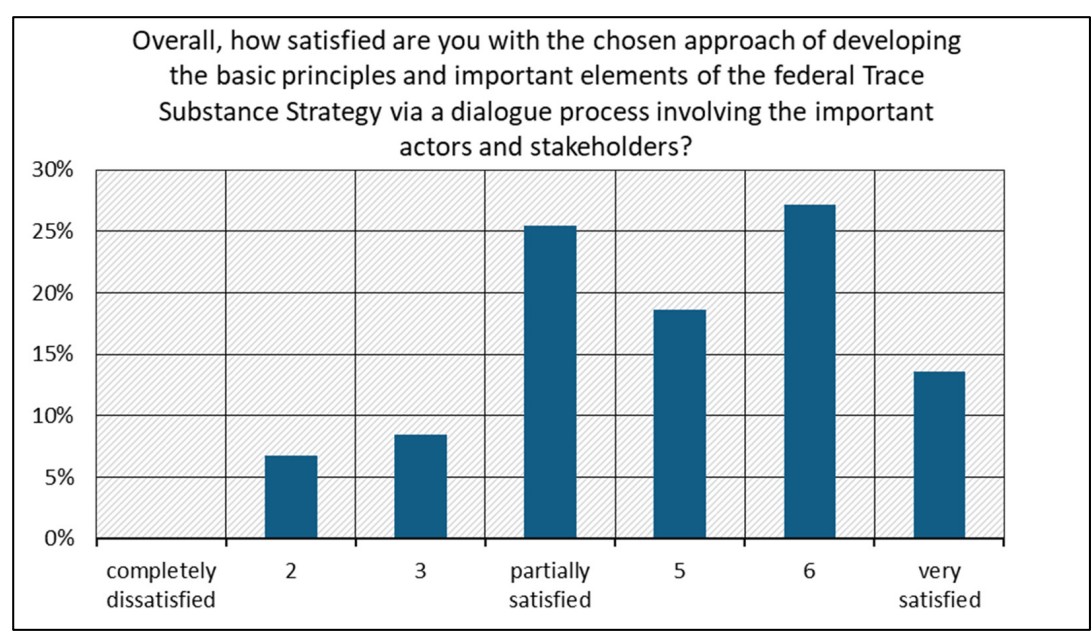

**Figure 11.** Survey results—satisfaction with the procedure. The graph is based on N = 53 responses. Source: Fraunhofer ISI.

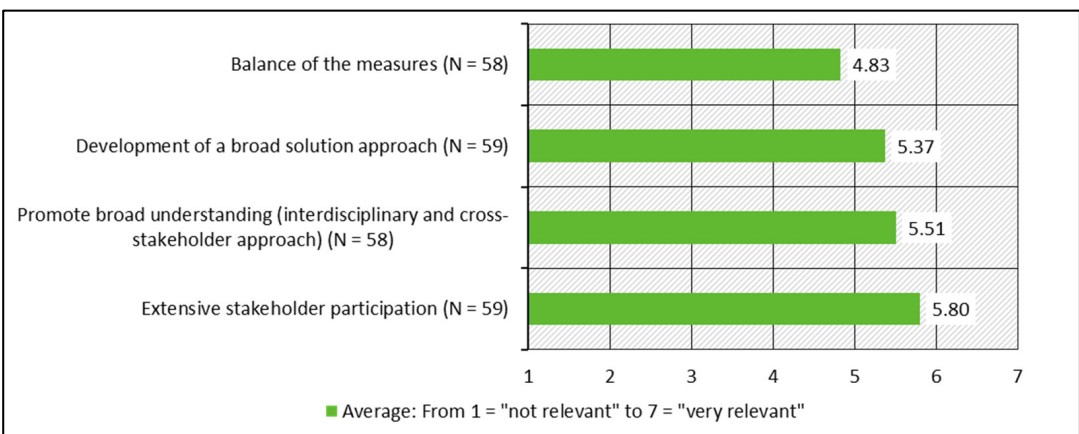

**Figure 12.** Survey results—relevance of the benefits of a dialogue process. Source: Fraunhofer ISI.

Across the board, stakeholders rated their satisfaction with the results of the two phases of stakeholder dialogue and the pilot phase at an average of 4.4 out of 7, i.e., a value between "partially satisfied" and "satisfied" (Figure 13). Differentiated by stakeholder groups, the federal states/environmental administration group shows the highest satisfaction (5.0), while the environmental associations/civil society group shows the lowest (3.43; Figure 14).

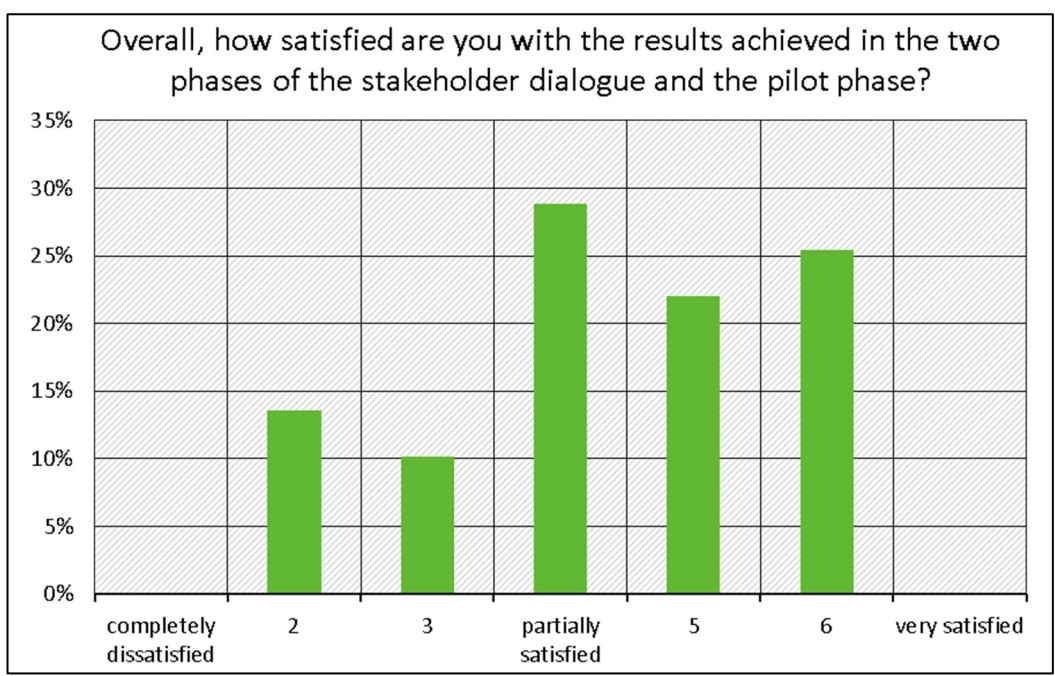

**Figure 13.** Survey results—satisfaction with the results. The graph is based on N = 53 responses. Source: Fraunhofer ISI.

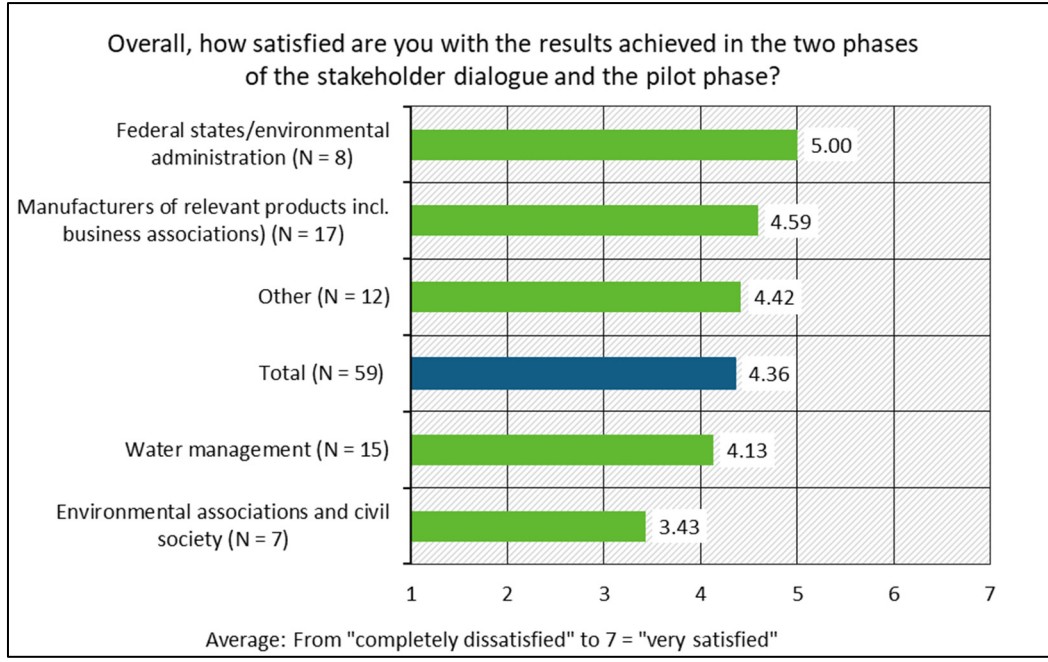

**Figure 14.** Survey results—satisfaction with the results, differentiated by stakeholder groups. Source: Fraunhofer ISI.

## 4. Discussion and Recommendations

Based on the Harvard concept [31] and the analyses of stakeholder dialogues in the environmental sector [32], the stakeholder dialogue on trace substances (micropollutants) was established as an instrument to develop the German government's Trace Substance Strategy and to enable the necessary policy mix with a view to the different sectors involved. This is in line with [15], who pointed out the power balance between stakeholders as an example of a means for ensuring sustainable outcomes from the development of policy mixes and overcoming conflicting and counterproductive effects. In their comparison of

305 case studies from 22 western democracies, Newig et al. [37] showed that the delegation of power to participants has a positive impact on environmental governance outcomes. Newig et al. [37] distinguished the three aspects of "intensity of communication among participants and process organizers," "extent to which participants can shape decisions" and "extent to which different stakeholder groups are represented" within a regression analysis. In this analysis, power delegation, which was an essential feature in the process described here, has shown to be the most stable predictor of strong environmental outputs. Thus, participation processes have become an important tool for enabling more environmentally effective decision-making.

As shown in Section 2.1, the multi-stakeholder dialogue on micropollutants initiated in 2016 greatly emphasized a strong integration of the stakeholders and enabled the stakeholders to develop the core elements of the German government's Trace Substance Strategy. Intense communication efforts were required to enable the stakeholders from various sectors to formulate the recommendations and measures that are expected for a relevant and strong environmental output. The environmental stance of all stakeholders was improved by the intensive communication amongst the participants, which helped to develop the overall positive outcomes. Overall, the outcomes fit very well into the different findings of literature.

Various activities within the framework of the German government's Traces Substance Strategy and the multi-stakeholder dialogue served as sources for articles on the governance of micropollutants, such as the integration of environmental organizations within the regulation of micropollutant emissions in water bodies [36] or the transition towards sustainable pharmacy [38].

While the stakeholder dialogue was used as an instrument, the important achievements from it were following measures:

- The implementation of The German Centre for Micropollutants (SZB)
- The implementation of an expert panel for final decision-making on the relevance of micropollutants after assessment by the UBA
- The establishment of roundtables for selected micropollutants to engage responsibility of industry
- Information campaign(s) under the umbrella of the UN Water Action Decade
- The application of the "orientation framework" for advanced wastewater treatment in the federal states
- Discussing the topic of funding.

It is important to understand that all stakeholders were involved in the dialogue process. However, the involvement in the implementation of the various measures varies between the stakeholder groups. The roundtables for selected micropollutants are the ones where the strongest involvement of the different stakeholders took place while, for example, the application of the orientation framework for advanced wastewater treatment in the federal states is targeting this specific group.

The available results from the overall evaluation show that the process itself and the chosen procedure (dialogue process) are positively rated overall by the stakeholders, while the group-specific ratings differ between the stakeholders, both with regard to the process and the results. A (slightly) positive assessment also prevails with regard to the results achieved, although this assessment varies depending on the stakeholder group, with the environmental associations and civil society group expressing slight dissatisfaction. When it comes to the results achieved, the high satisfaction levels of manufacturers and the lower satisfaction levels of the water management and environmental associations and civil society groups could be a sign that the stakeholder dialogue has been a successful instrument for preventing stricter regulation of micropollutants, from which manufacturers would benefit the most. However, it is important to consider that from the outset, the process aimed to push the relevant stakeholders to initiate important measures on a voluntary basis, including the discussion of market-based instruments. It was always clearly communicated that if these measures were not sufficient, regulatory measures would have

to follow. This may explain the high satisfaction levels expressed regarding the results on the side of the federal states/environmental administration (highest satisfaction of all stakeholder groups), seeing the process in a realistic way. However, no stakeholder group was completely satisfied or dissatisfied, which is quite normal since all outcomes represent a common compromise between all engaged stakeholders.

The new instruments and approaches developed as part of the dialogue process, some of which have already been established, have already gained a high level of acceptance across all stakeholder groups.

The results show that participation processes can be important and helpful for complex issues such as the field of water protection, especially as a means of proactively involving all stakeholders despite sometimes very different interests and prior knowledge. However, the effort involved is high and, compared to the development of new legal requirements, the overall process can only be accelerated to a limited extent. Since it is a complex overall situation, in the case of changes that are to be achieved in the long term, it is crucial that all groups involved are proactively included and encouraged to undertake their own activities.

When carrying out such a process, external professional support is absolutely necessary in order to at least partially compensate for the different starting conditions of the groups of actors (resources, level of information, etc.), e.g., through expert input and independent moderation (see [32], p. 47).

The results achieved within the framework of the dialogue process form the basis for the further design of the federal government's Trace Substance Strategy. In principle, the question of the additional measures demanded in some cases must be examined.

In 2018, the BMUV started the "National Water Dialogue" with the aim of establishing a "National Water Strategy" that would meet the challenges of climate and demographic change, aging water infrastructures and continuing water pollution (https://www.bmuv.de/en/topics/water-resources-waste/water-management/national-water-strategy accessed on 3 August 2023). It immediately became obvious that the results of the stakeholder dialogue on trace substances could be directly used within these discussions. They were integrated in the Draft National Water Strategy of the Environment ministry [39].

For the success of the overall process, it will be important in the future to set goals that are as specific as possible and to conduct an exchange about the achieved (interim) results with further participation of the stakeholders with a high degree of transparency.

Based on the feedback received during the evaluation activities, the following future tasks of the actors involved in the process can be noted as a guide:

- BMUV/UBA:

The German Centre for Micropollutants (SZB) will play a key role. In addition, questions of financing as well as overarching success monitoring have to be dealt with. It is also important to coordinate international activities in this field and to specify complementary measures.

- Product manufacturers/industry:

The essential elements here include the implementation of the agreements from the roundtables, the ongoing support of the roundtables as an instrument as well as the cooperation and acceptance of the work of the panel for the assessment of the relevance of trace substances/micropollutants.

- Federal states/environmental administration:

Implementing the "orientation framework" for extended wastewater treatment is a very important task for the German federal states. In addition, the federal states are responsible for designing and evaluating targeted monitoring programs on micropollutants. The environmental administration will also play an important role in future roundtable activities.

- Water management:

On the one hand, the water industry is a key player in the implementation of the results of the "orientation framework." The water industry also plays a decisive role in

the roundtable activities and in the implementation of awareness-raising measures and targeted monitoring programs.

- Environmental associations/civil society:

Environmental associations and civil society are also important actors in future roundtables. They can provide crucial support in the implementation of measures, e.g., with regard to raising public awareness.

The cross-sectoral importance of the stakeholder dialogue on micropollutants is underlined by the formal admission to the third National Action Plan within the Open Government Partnership (OGP). The key issues the dialogue must address in order to establish the combating of micropollutants as a long-term task are transparency, corporate social responsibility, data and information sharing, and the formulation of concrete measures on the basis of these activities.

The key findings from using a stakeholder dialogue to develop the German Trace Substance Strategy are as follows:

- A dialogue process is an important and helpful instrument for complex issues, especially when there are a large number of stakeholders with very different interests and prior knowledge.
- As a result, a new approach and new instruments were developed so that flexible and short-term options for action are now available.
- It also makes sense to conduct an evaluation of the environmental improvements that have actually been achieved and to consolidate the process, in parallel to the further implementation of the measures. This may result in the need for further developments and adjustments of measures and instruments.

**Author Contributions:** Conceptualization: T.H. and F.T.; Methodology: T.H., F.T. and M.B.; Validation: F.T., A.E., J.K. and J.R.; Formal analysis: T.H. and F.T.; Investigation: T.H. and F.T.; Resources: T.H., F.T. and M.B.; Writing—original draft: T.H., F.T., A.E. and J.R.; Visualization: T.H.; Writing—review & editing: T.H., F.T., M.B., S.L., A.E., J.K. and J.R.; Supervision: T.H., A.E. and J.R.; Project administration: T.H.; Funding acquisition: T.H., F.T. and M.B. All authors have read and agreed to the published version of the manuscript.

**Funding:** The organization, coordination and the scientific support of the different phases of the stakeholder dialogue on micropollutants was funded and supported by the German Federal Environment Agency (UBA) and the Federal Ministry for the Environment, Nature Conservation, Nuclear Safety and Consumer Protection (BMUV), funding code 3716 22 200.

**Data Availability Statement:** For more details, please see [33].

**Acknowledgments:** We would like to thank all stakeholders and the engaged participants for their contribution to the process of the stakeholder dialogue on micropollutants and for participating in its evaluation.

**Conflicts of Interest:** The authors declare that they have no competing financial interests or personal relationships that could have appeared to influence the work reported in this paper. The funding organizations were involved in the creation of the study design and accompanied the various stages of the process. The funding organizations supported the writing of this article and the decision to submit it for publication. The funding organizations were not involved in the collection, analysis or interpretation of the data.

## Appendix A

**Table A1.** List of involved stakeholders (during the process, which lasted several years, various changes occurred).

| |
|---|
| ABL—Arbeitsgemeinschaft Bäuerliche Landwirtschaft e.V. |
| Landesapothekerkammer Baden-Württemberg |
| BAG SELBSTHILFE |
| BAH—Bundesverband der Arzneimittel-Hersteller e.V. |
| BASF SE |
| Bayer AG |
| BBU—Bundesverband Bürgerinitiativen Umweltschutz e.V. |
| BDEW—Bundesverband der Energie- und Wasserwirtschaft e.V.<br>Niersverband für BDEW—Bundesverband der Energie- und Wasserwirtschaft e.V. |
| Currenta GmbH & Co. OHG für BDI—Bundesverband der Deutschen Industrie |
| BPI—Bundesverband der Pharmazeutischen Industrie e.V. |
| BUND—Bund für Umwelt und Naturschutz Deutschland |
| Umweltministerium Hessen für Bund-/Länderarbeitsgemeinschaft Wasser (LAWA)<br>Umweltministerium Nordrhein Westfalen (MULNV) für Bund-/Länderarbeitsgemeinschaft Wasser (LAWA)<br>Umweltministerium Baden-Württemberg für Bund-/Länderarbeitsgemeinschaft Wasser (LAWA) |
| Deutsche Krankenhausgesellschaft e.V. |
| Deutscher Bauernverband e.V. |
| Deutscher Landkreistag |
| Deutscher Städte- und Gemeindebund |
| Deutscher Städtetag<br>Stadtentwässerung Braunschweig für Deutscher Städtetag |
| DIHK—Deutscher Industrie- und Handelskammertag e.V. |
| DVGW—Deutscher Verein des Gas- und Wasserfaches e.V. |
| DWA—Deutsche Vereinigung für Wasserwirtschaft, Abwasser und Abfall e.V. |
| Gesamtverband der deutschen Textil- und Modeindustrie e. V. |
| IKW—Industrieverband Körperpflege- und Waschmittel e. V. |
| IVA—Industrieverband Agrar e.V.<br>Bayer AG für IVA—Industrieverband Agrar e.V. |
| Pro Generika e.V. |
| VCI—Verband der Chemischen Industrie e.V. |
| ver.di Vereinte Dienstleistungsgewerkschaft |
| Verbraucherzentrale Bundesverband e.V. vzbv |
| vfa—Verband forschender Arzneimittelhersteller |
| VKU—Verband kommunaler Unternehmen<br>Emschergenossenschaft/Lippeverband für VKU—Verband kommunaler Unternehmen |

Source: [35].

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
