# Peer review of "Engaging Stakeholders to Solve Complex Environmental Problems Using the Example of Micropollutants"

_water, doi:10.3390/w15193441_

Round 1

Reviewer 1 Report

Overall, this paper provides interesting insights into the design and implementation of a stakeholder participation process in environmental governance. The topic of micropollutants and how to govern the issue is of high relevance and the topic thus suits the journal well. Nevertheless, the paper is not ready for publication yet due to the following reasons: first, the research interest is not clear; second, information on the methodology for this particular paper is lacking (which directly relates to the first point); third, it is not clear how the reported findings related to the research interest (which could be solved by clarifying the first point and streamlining the results); fourth, the paper lacks integration into the literature on water governance and thus its contribution to literature is not clear. Therefore, I recommend publication of the article after major revision.

For further detailed comments and suggestions for revision, see below:

Abstract

I would recommend shortening the abstract and emphasizing the main points. At the moment, it is rather a description of the stakeholder dialogue than a paper abstract, which would include the research interest of this particular paper, its theoretical perspective / background, the research approach taken (design, methods) and empirical findings / conclusions.

Introduction

The introduction lacks a proper definition of what micropollutants or trace substances are.

“Micropollutants, in the context of the German multi-stakeholder dialogue referred to as trace substances, can have adverse effects on aquatic ecosystems even at low concentrations and affect the production of drinking water. The accumulation of poorly degradable substances, by-products that may be formed, or even the effects of the complex mixtures of substances that result in the environment are additional causes for concern.” These need references. For instance, Triebskorn et al. (2019).  

The issue is not only complex because of different sources of pollution but also because of the involvement of different political interests. See, for instance, Schaub & Braunbeck (2020) or Metz & Ingold (2017) Overall, the research interest is not clearly described in the introduction.

At the moment, the introduction lacks a clear statement, what the aim of the paper is and what overall research question it aims to answer. In its current form, it reads more like a report than a research article. Is it an evaluation? A research article on stakeholder participation in governance arrangements?

In addition, the paper lacks integration into existing literature, especially on water governance. There are several studies on the governance of micropollutants. Authors include Karin Ingold, Sabrina Kirschke, Florence Metz, Simon Schaub or Jale Tosun. Situating the paper into this literature would make clear, why this paper is of interest for the readership of Water and how its findings add to academic literature on the governance of micropollutants.  

Methods

“The introductory remarks (chapter 1) already indicate the overall very complex initial situation that exists in the subject area of limiting environmental pollution by micropollutants.” For a reader new to this issue, it is not quite clear why the issue is especially complex. But, see for instance Kirschke & Kosow (2022) or Kirschke et al. (2019).

“The present work showed that to develop the necessary comprehensive approach, "sectoral policies" are not sufficient; rather, cross-sectoral perspectives and approaches are needed aiming well beyond existing regulations. Therefore combinations of specific individual means are necessary and, at the same time, "established sectoral communication structures" must be overcome.” This sentence needs backing up. There is a literature on policy mixes that makes exactly this argument. See, for instance, Schaub, Vogeler & Metz (2022).

The methods section is mostly on the method of how the stakeholder dialogue was conducted. An essential missing part is information on the method how this particular paper assesses / investigates the stakeholder dialogue as a research item. The latter is a core part of a research article and needs to be explained. This point is strongly related to my previous point on research interest and research question.

Page 9: I would expect a bit more background information on the included actor groups (not only in the supplementary materials). How can they be classified? How equally balanced were different interests?

Figure 3: I recommend revising this figure. “Civil society” is quite a broad term and can involve all sort of interests. Further differentiation would be helpful here. What is meant by “Countries”? I assume the dialogue included only national representatives. Do you mean “Bundesländer”? In this case, “Federal States” is the appropriate term. What is meant by “Water res. man.”? What are “Local authorities”? “Producers” of what?

Page 15: “focus was therefore on an evaluation of the overall process and its sub-areas as part of the evaluation carried out in several steps during the pilot and continuation phases. The evaluations were essentially based on interviews with all stakeholders involved in the overall process and in the individual processes.” If this is the main focus of this paper, evaluating the process of the stakeholder dialogue, then this should be made clear right from the start (introduction). In addition, the methods section should make how exactly this evaluation was conducted (what are the criteria for evaluation? what methods were used? what kind of interviews? how was the survey conceptualized? what do the chosen survey questions measure and why were they chosen? how was the collected information analysed? qualitatively? quantitatively?). In addition, you state here that interviews were conducted. However, you report results from a survey in the findings section.

Findings

Figure 8: Not only is the approval rate by manufacturers lowest, the one by environmental associations and civil society is second lowest. Since these tend to have polarizing positions on environmental regulation, how would you explain this finding? “Other” is quite a large group. What actors does it include?

Page 20: The question on instrument use to develop and implement measures is not clear. The actors are very different with different competences. If I understand correctly, not every actor could implement every measure. Thus, more details and differentiation on this part is needed. Otherwise, the implications of these answers to the survey question are not clear. Moreover, the way I read the survey question Figure 9 is based on, it is more on whether actors generally believe that this instrument is useful. There, I am less surprised that water management and environmental associations are most doubtful since these are the actors who would prefer a strong state to just take regulatory action. See, for instance, Schaub & Tosun (2021) on the positioning of environmental groups in the stakeholder dialogue.

Page 22: “somewhat less relevance was subsequently given to the balance of measures (mean 4.83 out of 7; cf. Figure 12).” What exactly does this mean? What causes the imbalance in the view of different stakeholders?

Discussion

Page 24: “The available results show that the process itself and the chosen procedure (dialogue process) are positively rated overall by the stakeholders.” This statement is not backed by the survey results. They rather show that satisfaction differs between the stakeholder groups. Both with regard to the process and the results.

It is apparent that manufacturers show highest satisfaction with the stakeholder dialogue whereas it is lowest with water management and environmental associations (the two actors most strongly in favor of strict regulation). Moreover, most of the agreed measures are voluntary measures and information campaigns. Regulatory or market-based instruments are largely missing. Has the stakeholder dialogue been a successful format for manufacturers to prevent stricter regulation of micropollutants? When evaluating the stakeholder dialogue, I would expect some sort of discussion on this question.

Typically, the discussion section compares a paper’s findings with those of existent literature. Are the findings different to what has been reported previously? Do they point to new interesting avenues for research? In this particular case: How do the insights gained on the stakeholder dialogue relate to experiences with other similar processes of stakeholder participation? Are there any findings in literature with which these findings can be compared with? For an overview on stakeholder participation in environmental governance, see Newig et al. (2023).

Minor points

Some terms need clarification to enhance readability. For instance:

-        “environmental status of selected pollutants” on page 5, lines 155f.

-        precautionary and polluter-pays principles

Conflict of interests

The authors should make transparent their role in the design and implementation of the stakeholder dialogue and declare how they have ensured that their evaluation is non-biased.

Literature

Kirschke, S., & Kosow, H. (2022). Designing policy mixes for emerging wicked problems. The case of pharmaceutical residues in freshwaters. Journal of Environmental Policy & Planning, 24(5), 486-497.

Kirschke, S., Franke, C., Newig, J., & Borchardt, D. (2019). Clusters of water governance problems and their effects on policy delivery. Policy and Society, 38(2), 255-277.

Newig, J., Jager, N. W., Challies, E., & Kochskämper, E. (2023). Does stakeholder participation improve environmental governance? Evidence from a meta-analysis of 305 case studies. Global Environmental Change, 82, 102705.

Schaub, S., & Braunbeck, T. (2020). Transition towards sustainable pharmacy? The influence of public debates on policy responses to pharmaceutical contaminants in water. Environmental Sciences Europe, 32, 1-19.

Schaub, S., & Tosun, J. (2021). Politikgestaltung im Dialog? Umweltgruppen und ihre Mitwirkung bei der Regulierung von Spurenstoffen in Gewässern. ZPol Zeitschrift für Politikwissenschaft, 31(2), 291-325.

Schaub, S., Vogeler, C., & Metz, F. (2022). Designing policy mixes for the sustainable management of water resources. Journal of Environmental Policy & Planning, 24(5), 463-471.

Triebskorn, R., Blaha, L., Gallert, C., Giebner, S., Hetzenauer, H., Köhler, H. R., ... & Wilhelm, S. (2019). Freshwater ecosystems profit from activated carbon-based wastewater treatment across various levels of biological organisation in a short timeframe. Environmental Sciences Europe, 31(1), 1-16.

mostly fine, only minor issues. I recommend proof-reading by a native speaker before publication.

Reviewer 2 Report

In this manuscript, the authors described that German Government developed a trace substance strategy in 2016 to address challenges such as the climate crisis and achieve the objectives of the Water Framework Directive through a holistic and precautionary approach. This strategy involves stakeholder dialogue, development of measures, and the establishment of instruments to effectively reduce micropollutant emissions in the water sector, including advanced wastewater treatment. The manuscript is well written, but the authors must address some concerns before it is considered for publication.

1) Line 282 – 232, 308 – 310, 575 – 576, please make this part’s template consistent with the rest of this manuscript.

2) Line 292, please delete this extra blank page.

3) Line 431, please correct the Table 1 title by deleting “Error! Reference source not found”. Also, please make the font size in Table 1 smaller.

4) Please pay more attention to the manuscript organization by making full use of the blank spaces such as Line 316 – 317 and 580 – 581.

5) Please make all figures (especially Figures 3 and 4) in this manuscript as clear as Figure 1.
